# Identification and Characterization of a Novel Endo-β-1,4-Xylanase from *Streptomyces* sp. T7 and Its Application in Xylo-Oligosaccharide Production

**DOI:** 10.3390/molecules27082516

**Published:** 2022-04-13

**Authors:** Yumei Li, Xinxin Zhang, Chunwen Lu, Peng Lu, Chongxu Yin, Zhengmao Ye, Zhaosong Huang

**Affiliations:** Department of Biological Science and Biotechnology, University of Jinan, Jinan 250022, China; mls_liym@ujn.edu.cn (Y.L.); bernice0605@163.com (X.Z.); lucw123456@163.com (C.L.); 201921200894@mail.ujn.edu.cn (P.L.); chauncey4u@126.com (C.Y.); mse_yezm@ujn.edu.cn (Z.Y.)

**Keywords:** endo-β-1,4-xylanase, expression, xylo-oligosaccharides

## Abstract

A xylanase-producing strain, identified as *Streptomyces* sp. T7, was isolated from soil by our lab. The endo-β-1,4-xylanase (*xyn*ST7) gene was found in the genome sequence of strain T7, which was cloned and expressed in *Escherichia coli*. XynST7 belonged to the glycoside hydrolase family 10, with a molecular mass of approximately 47 kDa. The optimum pH and temperature of XynST7 were pH 6.0 and 60 °C, respectively, and it showed wide pH and temperature adaptability and stability, retaining more than half of its enzyme activity between pH 5.0 and 11.0 below 80 °C. XynST7 showed only endo-β-1,4-xylanase activity without cellulase- or β-xylosidase activity, and it showed maximal hydrolysis for corncob xylan in all the test substrates. Then, XynST7 was used for the production of xylo-oligosaccharides (XOSs) by hydrolyzing xylan extracted from raw corncobs. The maximum yield of the XOS was 8.61 ± 0.13 mg/mL using 15 U/mL of XynST7 and 1.5% corncob xylan after 10 h of incubation at 60 °C. The resulting hydrolysate products mainly consisted of xylobiose and xylotriose. These data indicated that XynST7 might by a promising tool for various industrial applications.

## 1. Introduction

The plant cell wall is composed of lignin (5–15%), cellulose (37–50%) and hemicellulose (25–50%) [1]. Xylan is the major component of hemicellulose, as a cross linker between different components. Xylan is a heteropolysaccharide containing β-1,4-glycosidic linkages of the d-xylopyranoside backbone and complex side chains with arabinosyl, acetyl and glucuronosyl groups [2]. Hence, the complete enzymatic hydrolysis of complex polymers requires an array of enzymes with diverse action modes. 

Endo-β-1,4-xylanase (EC 3.2.1.8) is a key enzyme of xylanolysis, which results in changes of polymeric xylan into a mixture of xylo-oligosaccharides (XOS) [3]. Based on amino acid sequence similarities, endo-β-1,4-xylanases are classified into the glycoside hydrolase (GH) families 5, 7, 8, 10, 11, 16, 26, 30, 43, 52 and 62. GH16, 51 and 62 families appear to be bifunctional enzymes containing two catalytic domains, while GH5, 7, 8, 10, 11 and 43 families have a truly different catalytic domain with endo-1,4-β-xylanase activity [4]. The GH5 family is the largest glycoside hydrolase family, and only seven amino acid residues are strictly conserved among all of the members [5]. The GH8 family contains some cold-adapted xylanases [6]. On the basis of hydrophobic cluster analysis of the catalytic domains, xylanases have been mainly categorized as GH10 and 11 families [7]. The members of the GH10 family have higher molecular weights (>30 kDa) than those of the GH11 family, with acidic pIs and (β/α) barrel folds in their three-dimensional (3D) structure. The GH 11 family xylanases show lower molecular weights (<30 kDa) and basic pIs, and they consist of two large β-pleated sheets and a single a-helix that forms a structure similar to a partially closed right hand [8]. 

Xylanases are used widely in the food, animal feed, paper and pulping, and biofuel industries [9]. Studies conducted over the years have shown that enzymatic hydrolysis of xylan to generate XOS is one important application of xylanase [10]. The enzymatic production of XOS is greatly beneficial because it does not produce considerable harmful by-products or monosaccharides. XOS have various biological functions, for instance improving the growth of beneficial bacteria (*bifdobacteria* and *lactobacilli*), promoting calcium absorption, reducing the risk of colon cancer, enhancing immune-modulatory and anti-infective blood and skin effects, and inhibiting microbial actions [11]. Therefore, they have many essential medicinal uses. 

Xylanases are abundantly present in nature, arising both in prokaryotes and eukaryotes, and a large variety of them have been reported in microorganisms, including bacteria and fungi from various habitats. Among the prokaryotes, bacterial genera including various actinomycete species from *Streptomyces*, *Nonomuraea*, and *Actinomadura* are also reported for xylanase production [12,13,14,15]. Some *Streptomyces* species have been reported to produce thermostable xylanases active in the range of 60 to 70 °C [13,16]. It is well-known that the xylanases working at higher temperatures and pHs work in favor of several biotechnological applications. For instance, xylan hydrolysis facilitates the release of XOS from corncobs, while the use of thermo-alkaline xylanases allows direct extraction of xylan from raw corncobs using the mild alkali method, which avoids the need for temperature and pH readjustments and saves time and costs [17].

In this study, a strain producing xylanase was isolated from soil, which was identified as *Streptomyces* sp. T7. An endo-β-1,4-xylanase (*xyn*ST7) gene from strain T7 was cloned and expressed in *E. coli*, and the recombinant enzyme was characterized and used to produce XOS through hydrolyzing corncob xylan.

## 2. Results and Discussion

### 2.1. Obtaining Xylanase Secreting Strains

Five extracellular xylanase-producing strains were detected via enzyme activity assay. Among them, strain T7 showed the highest activity toward catalyze beechwood xylan hydrolysis. Based on the 16s rRNA sequence analysis, the target strain was identified as *Streptomyces* sp. T7, which possessed high similarity (>99.0%) to other strains of *Streptomyces* (Appendix A). As far as we know, xylanase produced by *Streptomyces* has excellent industrial development potential.

### 2.2. Sequence and Structure Analysis

The genomic DNA of *Streptomyces* sp. T7 was sequenced and submitted to GenBank database by our lab (GenBank accession No. JAKNRM010000000). An endo-β-1,4-xylanase gene was found, which encoded 460 amino acids including a 25 residue signal sequence (Appendix A). After processing of the signal peptide, the mature polypeptide contained 435 residues with a theoretical molecular mass of 46.6 kDa. Based on the comparative analysis against the NCBI Conserved Domain Database (CDD), the deduced amino acid sequence of XynST7 contained the conserved GH10 catalytic domain and a ricin-type lectin domain (Appendix A). As shown in Appendix A, the overall sequence of XynST7 showed high identity to the endo-β-1,4-xylanase, which has been experimentally characterized from *Streptomycesalbidoflavus* (99.78%, WP_047467668.1) and *Streptomyces viridochromogenes* (89.91%, WP_004003025.1) by BLASTp against the Refsep_protein database [18]. The sequence also showed substantially high identity to the endo-β-1,4-xylanase with 3D structure data from *S treptomyces olivaceoviridis* E-86 (91.26%, 94% coverage; 1ISV_A: 2-436), *Streptomyces lividans* (91.33%, 65% coverage; 1E0W_A: 2-301), *Streptomyces halstedii* JM8 (61.51%, 62% coverage; 1NQ6_A: 13-302), *Streptomyces* sp. 9 (44.44%, 64% coverage; 3WUB_A: 3-313), and *Cellulomonas flavigena* (63.79%, 64% coverage; 5M0K_A: 39-336). Moreover, the XynST7 showed 88.51% identity to XlnA-GH10 from *Streptomyces* lividans by BLASTp against the UniProtKB database. XlnA-GH10 is a reported modular protein that possesses typical endo-β-1,4-xylanase activity and cellulase-free activity [19]. The multiple alignment was carried out using those sequences with structural information with the ClustalW tool, whereby two putative catalytic site residues, E127 and E235, were found in the conserved sequence motifs WDVVNEAF and TELDI of XynST7, respectively (Figure 1) [20,21].

The homology model of XynST7 was generated using the crystal structures (1ISV_A) of GH10 endo-β-1,4-xylanase from *S. olivaceoviridis* as the template, which indicated that the mature enzyme had a classical GH10 family framework. The putative overall structure of XynST7 was composed of three parts: a catalytic domain with a conserved (β/α)8-barrel topology in the *N*-terminal end (residues 1–301), in which the active site loops are arranged to form a deep cleft; an 11-residue Gly/Pro-rich linker region in the middle (residues 302–312), and a ricin-type lectin domain belonging to CBM family 13 that was formerly a xylan-binding domain (residues 313–435) in the C- terminal end. As expected, two predicted catalytic residues, E127 (acid/base) and E235 (nucleophile), were located within the active site cleft in the N-terminal end, which was similar to other GH10 endo-β-1,4-xylanases (Figure 2a) [20]. Furthermore, the residues of Glu45, Asp271, Arg274, Gln 88, Trp265, and Trp273 in the active site of XynST7 were predicted to participate in the catalytic reaction or substrate binding using 1ISV_A as a template (Figure 2b). These highly conserved residues were connected to the catalytic mechanism of GH10 xylanases, which was inconsistent with the sugar-binding modes of SoXyn10A in the catalytic cleft that was previously reported to have substrate specificity toward arabinoglucuronoxylan [21]. Furthermore, the natural XynST7 was presumed to be a symmetrical dimer according to the homo-dimeric template of endo-β-1,4-xylanase (Figure 2c), which was observed in a previous study [22].

### 2.3. Expression and Purification 

Recombinant endo-β-1,4-xylanase was abundantly expressed in *E. coli* BL21 (DE3). The recombinant protein was approximately 47 kDa, which was in agreement with the calculated molecular mass (46.6 kDa) of XynST7 (Figure 3) and in the range (>30 kDa) of the great majority of GH10 xylanases [4,8,9].

### 2.4. Biochemical Characteristics

The XynST7 was active between pH 5.0 and 11.0, showing maximum activity at pH 6.0, while XynST7 was able to retain more than 60% of the residual activity in the range from pH 5.0 to pH 11.0 after 24 h of incubation at 4 °C (Figure 4). The pH stability of several xylanases from *Streptomyces* sp. S9 and *Streptomyces* sp. TN119 has been reported to be between pH 5.0 and 11.0 [16,23]. Moreover, the majority of xylanases from *Streptomyces* genus were active and stable between pH 5.0 and 9.0 [13,16]. In addition, the previous study showed that proteins with charged surfaces, skewing either to negative or positive charges, are good for extreme pH environments [24]. The optimum pH of the xylanase activity appeared to increase with the ratio of negatively to positively charged residues. Based on the total amino acid compositions of the catalytic domains that were calculated using the ExPASy–ProtParam Tool, XynST7 is acidic and has a high ratio of negatively (Asp and Glu) to positively (Arg and Lys) charged residues (Appendix A), which are similar to other alkaline active xylanases [24]. This is also reflected in the surface electrostatic potential of XynST7 (Appendix A).

The XynST7 exhibited the maximum activity at 60 °C, which was in agreement with certain xylanases from *Streptomyces* sp. S9 (XynAS9), *S. halstedii* JM8 (Xys1L), *S. fradiae* var. k11 (SfXyn10), and *A. terreus* (BCC129) [16,25,26,27]. The enzyme showed approximately 60% of the maximum activity when it was assayed between 40 and 70 °C, while more than 40% of the enzyme activity was detected below 20 °C or above 80 °C (Figure 5a). Likewise, XynST7 showed excellent thermostability, retaining over 90% of the residual activity after 60 min of incubation at 60 °C. Furthermore, XynST7 still retained more than 40% of the residual activity when it was incubated at 80 °C for 60 min (Figure 5b). Under the same conditions, some xylanases were also active, such as the aforementioned XynAS9, Xys1L, SfXyn10 and BCC129. Although the thermostable xylanase reported previously had the highest activity at 80 °C, its half-life at 80 °C was less than 10 min. Another thermostable xylanase from a thermophilic strain showed maximum activity at 70 °C, although the enzyme activity drastically decreased at more than 65 °C and retained only 5% of its activity after preincubation at 70 °C for 10 min [28]. Thus, XynST7 showed good activity and stability over a wide range of temperatures, which has made it a potential candidate for various industrial processes. 

The effects of metal ions and reagents on the activity of XynST7 are shown in Table 1. The enzyme activity was substantially enhanced by Mn^2+^, which is similar to other endo-β-1,4-xylanases from *Streptomyces thermocarboxydus* HY-15 and *Cellulosimicrobium* sp. HY-12 [29,30]. The enzyme activity increased at low concentrations of K^+^, Co^2+^, and Mg^2+^ (≤1 mM), while a drastic inhibition was observed in the presence of Fe^3+^, Fe^2+^, SDS, and EDTA in a concentration-dependent manner, in agreement with the data from *Geobacillus thermodenitrificans* A333, except for EDTA [31]. Little effect was found in the presence of Na^+^, Ca^2+^, and Ni^2+^, as reported in the xylanase from *Streptomyces* sp. 9 [16].

### 2.5. Substrate Specificity and Kinetic Properties

The hydrolytic properties of XynST7 toward natural and artificial substrates are shown in Table 2. XynST7 decomposed beechwood, birchwood, corncob, and oat spelt xylans with different specificities, which is characteristic of xylanases belonging to GH10 glycosyl hydrolases [8]. When compared to xylan hydrolysis (≥90% xylose) of beechwood, birchwood, corncob, and oat spelt xylans, the specific activity of xXynST7 for corncob xylan was determined to be approximately 235.8 U/mL, which was slightly higher than those of beechwood xylan (203.3 U/mL), oat spelt xylan (193.3 U/mL), and beechwood xylan (188.6 U/mL). Therefore, the susceptibilities of xylosic materials to XynST7 were listed as follows: corncob xylan > birchwood xylan > oat spelt xylan > beechwood xylan. The results were similar to the substrate specificity from *S. thermocarboxydus* HY-15 and *Streptomyces* sp. S38 [29,32]. The XynST7 is not active towards either amorphous or crystalline cellulose represented by carboxymethylcellulose and Avicel, suggesting that it is a cellulase-free xylanase. Moreover, the activity of XynST7 against *p*-nitrophenyl-saccharide derivatives (*p*NPA, *p*NPX, *p*NPGα, and *p*NPGβ) was negligible, meaning no α-arabinofuranosidase, β-xylosidase, or α- or β-glucosidase activities were detected. These results clearly highlight the strict specificity of XynST7 toward polysaccharides containing β-1,4-xylopyranose units, which showed the monospecific characteristics of GH10 xylanases [33]. Thus, XynST7 was applicable to XOS production via hydrolyzing corncob, since less or negligible amounts of exo-xylanase or β-xylosidase activity can contribute to high amounts of xylose, which may cause inhibitory effects on XOS production. 

In this case, the kinetic parameters of XynST7, including *K*m, *V*max, and *k*cat values for various xylans, were evaluated. The XynST7 showed the highest *V*max value (approximately 678.14 U/mg) and lowest *K*m value (approximately 1.33 mg/mL) toward corncob xylan. Additionally, the respective *V*max and *K*m values of XynST7 were 596.54 U/mg and 2.78 mg/mL for birchwood xylan, 545.33 U/mg and 3.12 mg/mL for oat spelt xylan, and 495.23 U/mg and 3.54 mg/mL for beechwood xylan, respectively. The catalytic efficiency levels (*k*cat) of XynST7 for corncob xylan were higher by approximately 1.36-, 1.67-, and 2.08-fold than for birchwood, oat spelt, and beechwood xylan, respectively. Taken together, these findings indicate that XynST7 is a true endo-β-1,4-xylanase that exclusively cleaves the β-1,4-xylosidic linkages in different xylosic compounds.

### 2.6. XOS Production

Xylan from the mild alkali extraction of raw corncobs was used for enzymatic hydrolysis to produce XOSs. As shown in Figure 6a, it seems adequate for achieving maximum production of XOSs in a hydrolysis period of 6–10 h. The yield of XOS increased to 7.76 ± 0.2 mg/mL after 6 h of incubation. The production of XOS slowly increased after 6 h, which only increased up to 1.24 ± 0.15 mg/mL with the yield of 7.99 ± 0.1 mg/mL, even after 10 h of incubation. This phenomenon may be due to the decreased level of easily accessible hydrolytic sites in the xylan chain or decreased endo-xylanase activity due to end product inhibition. Chapla et al., previously reported a best time period of 8–16 h for XOS production from corncob xylan using β-xylosidase-free xylanase from *Aspergillus foetidus* MTCC 4898 [34]. Akpinar et al., reported that 8–24 h was the optimum time for the production of XOS through enzymatic hydrolysis of agricultural wastes using commercial xylanase [35]. The present XOS production exhibited a distinct advantage in terms of the shorter reaction time, i.e., 6–10 h, with the yield of XOS being up to 7.99 ± 0.1 mg/mL without the generation of xylose. Figure 6b shows that the yield of XOS increased from 4.31 ± 0.16 to 8.52 ± 0.21 mg/mL with xylanase dosages from 5 to 30 U/mL, while the optimum xylanase concentration should be 15 U/mL. The substrate concentration also plays an important role in the process of enzymatic hydrolysis. A substantial increase in XOS production appeared with increasing substrate concentrations from 1.5% to 3%, but then drastically decreased the yield of XOS with decreasing the concentration of corncob xylan in the reaction mixture from 1.5% to 0.3%. The maximum yield of XOS was 8.61 ± 0.13 mg/mL after 10 h of incubation using 1.5% of corncob xylan as the substrate (Figure 6c). This may have been due to a reduction in water content in the reaction system with the higher substrate concentration resulting in a reduction in XOS production. The temperature was an essential factor for the enhancement of XOS production, while the maximum yield of XOS was observed at 60 °C after 10 h of incubation (Figure 6d). The production of XOS was retarded below or above 60 °C owing to the disadvantages of the enzyme reaction at lower temperatures or inactivation of enzymes at higher temperatures during longer run times.

XynST7 has a greater advantage for XOS production over other xylanases from fungi, such as *Penicillium funiculosum*, *Jonesia denitrificans,* and *Aspergillus fumigatus* var. niveu, as well as over bacteria such as *Thermobacillus composti*, *Paludibacter propionicigenes,* and *Streptomyces* sp. B6 [36,37,38,39,40,41]. XynST7 has better pH stability and thermostability than others, while XynST7 utilized xylan to produce XOS and no xylose, which may be of benefit during XOS purification (Table 3).

Based on these results, XOS production was performed using 15 U/mL of XynST7 and 1.5% corncob xylan at 60 °C incubation for 24 h. The resulting XOSs (from 6, 10, 24 h) were analyzed via TLC. The most noteworthy was that only XOSs (xylobiose and xylotriose) were observed during the process of XOS production (Figure 7a), meaning it would be beneficial to separate XOSs in the absence of xylose formation, which might make the production process more cost -effective and economic. It was further confirmed that XOS mainly consisted of xylobiose (X2) and xylotriose (X3) via HPLC (Figure 7b). It is well known that xylobiose and xylotriose are considerably important prebiotics.

## 3. Materials and Methods 

### 3.1. Strains, Plasmids, and Chemicals

*E. coli* DH5α and pMD 18-T vectors for gene cloning and *E. coli* BL21 (DE3) and pET-28a (+) for gene expression of endo-β-1,4-xylanase were preserved in our laboratory. Restriction endonucleases *Bam*H I and *Xho* I, PrimeSTAR^®^ HS DNA polymerase, and T4 DNA ligase were purchased from TaKaRa (Tokyo, Japan). The substrates *p*NP-α-d-arabinofuranoside (*p*NPA), *p*NP-β-d-xylopyranoside (*p*NPX), *p*NP-α-d-glucopyranoside (*p*NPGα), *p*NP-β-d-glucopyranoside (*p*NPGβ), carboxymethyl cellulose (CMC), and Avicel, as well as xylans from beechwood, birchwood, oat spelt, and corncob, were purchased from Sigma Chemical Company (St. Louis, MO, USA). All other chemicals used in the present study were of analytical grade.

### 3.2. Microorganism Isolation and Identification

Soil samples from the Colorful Desert (Tibet, China) were collected and used for the enriched culture with liquid medium containing 1% beechwood xylan, 3% yeast extract, 0.5% KH_2_PO_4_, 0.5% K_2_HPO_4_, and 0.5% MgSO_4_·7H_2_O. Subsequently, the microorganisms were gradually diluted with normal saline and screened using a primary screening medium (1% beechwood xylan, 5% yeast extract, 0.5% KH_2_PO_4_, 0.5% K_2_HPO_4_ and 1% MgSO_4_·7H_2_O, 2% agar). The xylan-producing colonies were screened through Congo Red staining and cultured on the fermentation medium (1% beechwood xylan, 2% KNO_3_, 2% KH_2_PO_4_, 0.5% MgSO_4_·7H_2_O, and 0.01% FeSO_4_·7H_2_O). The hydrolysis activity of the fermentation supernatant was analyzed via thin-layer chromatography (TLC). The strain with the highest activity for xylan hydrolysis was identified based on the analysis of the 16S rRNA sequence, which was amplified using primers 27F (5′-AGAGTTTGATCCTGGCTCAG-3′) and 1492R (5′-GGTTACCTTGTTACGACTT-3′).

### 3.3. Sequence and Structure Analysis

The presence of signal peptides was detected using the SignalP 6.0 server (https://services.healthtech.dtu.dk/service.php?SignalP-6.0, accessed on 3 January 2022). Protein homology searches were conducted using BLASTX (https://blast.ncbi.nlm.nih.gov/Blast.cgi, accessed on 3 March 2022). The multiple alignment was completed online using ClustalW (https://www.genome.jp/tools-bin/clustalw, accessed on 1 July 2014) and ESPrist 3.0 (https://espript.ibcp.fr/ESPript/cgi-bin/ESPript.cgi, accessed on 21 April 2014). Conserved domains were analyzed using National Center for Biotechnology Information (NCBI) CD-Search (https://www.ncbi.nlm.nih.gov/Structure/cdd/wrpsb.cgi, accessed on 3 March 2022). The classification of enzymes into GH families was carried out using InterPro (http://www.ebi.ac.uk/interpro, accessed on 10 March 2022). The theoretical molecular mass of the recombinant protein was predicted using the ExPASy ProtParam tool (https://web.expasy.org/protparam/, accessed on 2 July 2021). The three-dimensional structure of xylanase from *Streptomyces* sp. T7 (XynST7) was predicted with a Swiss-Model server (https://swissmodel.expasy.org/, accessed on 21 May 2018) using the xylanase from *S. olivaceoviridis* E-86 (PDB-ID: 1ISV) as the template and optimized based on the energy minimization.

### 3.4. Cloning and Recombinant Expression

The XynST7-encoding gene (removing the signal peptide sequence) was amplified using the forward primer (5′-ATGGATCCCAGAGCGGCCGCTACTTCGGCAC-3′ with *Bam*H I restriction site) and reverse primer (5′-ATCTCGAGTCAGGTGCGGGTCCAGCGCTG-3′ with *Xho* I restriction site), and ligated into the pMD 18-T vector. The resulting pMD 18-T-*xyn*ST7 was transformed into *E. coli* DH5α and sequenced by Tsingke Biotechnology Co., Ltd. (Qindao, China). The plasmid pMD 18-T-*xyn*ST7 was digested with BamHI and XhoI, and the resulting *xyn*ST7 was ligated to the pET-28a (+), which was transformed into *E. coli* BL21 (DE3). A single colony harboring the plasmid pET-28a-*xyn*ST7 was grown at 37 °C for 12 h in 3 mL of Luria–Bertani (LB) medium containing 50 µg/mL kanamycin. This culture was transferred to a fresh LB medium containing kanamycin according to the inoculation amount of 2% (*v*/***v***) and grown at 37 °C with shaking at 200 rpm. When the OD_600nm_ value of the culture reached 0.6, *xyn*ST7 expression was induced by adding isopropyl-β-D-thiogalactopyranoside to a final concentration of 1 mM, then cultivation was continued for another 3 h at 37 °C. The *xyn*ST7 expression was confirmed by 10% sodium dodecyl sulfate–polyacrylamide gel electrophoresis (SDS-PAGE).

### 3.5. Purification 

XynST7 was purified via a series of steps, including ultrasonication, Ni^2+^–nitrilotriacetate affinity chromatography, dialysis, and ultrafiltration. The *E. coli* BL21 (DE3) cells with *xyn*ST7 recombinant expression were disrupted by ultrasonication on ice (work time: 15 min; work/interval time: 3 s/5 s; ultrasonic output power: 200 W). The supernatant was collected by centrifugation at 12,000× *g* for 20 min at 4 °C, and applied to a Ni Sepharose 6FF column following the manufacturer’s protocols (Solarbio, Jinan, China). Protein fractions were eluted with 10 mM imidazole buffer first, then 100 mM imidazole buffer at a flow rate of 1ml/min. After dialysis for desalting, the collecting liquid was concentrated by ultrafiltration using a Millipore Ultrafiltration System with a molecular weight cutoff at 10 kDa (Millipore, USA). The purified protein was analyzed by SDS-PAGE. The molecular mass standards of protein were used in the range of 20–150 kDa (Thermo Scientific, Waltham, MA, USA).

### 3.6. Enzyme Activity Assay

Xylanase activity was assayed using beechwood xylan as a substrate. Here, 0.3 mL of 1% xylan in 0.1 M PBS buffer (NaH_2_PO_4_–Na_2_HPO_4_ buffer, pH 6.5) was incubated with 0.1 mL of enzyme at 60 °C for 30 min. The reaction was stopped using 0.6 mL of 3,5-dinitrosalicylic acid (DNS) reagent [42]. The vials were put in a boiling water bath for 10 min. The absorbance was measured at 540 nm after cooling. Reducing sugars released from the reactions were measured using a xylose standard curve. The total protein concentration was evaluated using a BCA Protein Assay Kit (Sangon Biotech, Shanghai, China) with bovine serum albumin as the standard. One unit (U/mL) of xylanase activity was defined as the amount of enzyme catalyzing the release of 1 mg/min of reducing sugar equivalent to xylose under the specified assay condition. Cellulase activity was determined using the same method as xylanase. Here, α-arabinofuranosidase, β-xylosidase, and α- and β-glucosidase were assayed with *p*NPA, *p*NPX, *p*NPGα, and *p*NPGβ as substrates, respectively, and their reaction mixtures were incubated at 60 °C for 10 min, respectively. The absorbance of the released p-nitrophenol at 410 nm was determined. 

### 3.7. Biochemical and Kinetic Characteristics 

For optimum pH, the enzyme activity was measured at various pH values (pH4.0 –11.0) using CH_3_COONa-CH_3_COOH buffer (pH 4.0–6.0), Na_2_HPO_4_-NaH_2_PO_4_ buffer (pH 6.0–8.0), and Gly-NaOH buffer (pH 8.0–11.0)) at 60 °C for 30 min. Under stable pH conditions, the enzyme was pre-incubated at various pH values (pH4.0–11.0) at 4 °C for 24 h. For the optimum temperature, the enzyme activity was measured at various temperatures (20–80 °C) and pH 6.5 for 30 min. Under stable temperature conditions, the enzyme was pre-incubated at various temperatures (20–80 °C) with 5–60 min of incubation at pH 6.5. The effects of different metal ions and additives on enzyme activity were determined by incubating the enzyme with solution containing Na^+^, K^+^, Ca^2+^, Fe^2+^, Mg^2+^, Mn^2+^, Co^2+^, Ni^2+^, Fe^3+^, EDTA, and SDS for 24 h at 4 °C. Kinetic parameters were determined by performing enzymatic reactions at 60 °C, with various xylans (0.8–4 mg/mL) in 0.1 M NaH_2_PO_4_–Na_2_HPO_4_ buffer (pH 6.5) as the substrate. The catalytic constant (*k*_cat_) and catalytic efficiency (*k*_cat_/*K*_m_) were calculated using *K*_m_ and *V*_max_ values determined from the Lineweaver–Burk plot.

### 3.8. Production and Analysis of XOS

Corncob xylan was prepared using a mild alkali extraction method, as previously described [43]. The raw corncobs (50 g) were blended with 800 mL of 1.25 M NaOH for 15 min and shaken for 3 h with 150 rpm at 40 °C, then centrifuged at 10,000× *g* for 20 min. The resulting supernatant was acidified to pH 5.0 with HCl, which was used as a substrate for enzymatic hydrolysis. The effects of various physico-chemical parameters, including the reaction time (1, 2, 3, 4, 5, 6, 7, 8, 9, 10, 12, 14, 16, 18, 20, 24 h), enzyme dosage (5, 10, 15, 20, 30 U/mL), substrate concentration (0.3%, 0.5%, 1%, 1.5%, 2% and 3%), and incubation temperature (45, 50, 55, 60, 65 °C), on the yield of XOS were successively investigated in order to maximize XOS production. Finally, XOS production was carried out by hydrolyzing corncob xylan under the optimum conditions. The resulting hydrolysates of corncob xylan were analyzed via thin-layer chromatography (TLC) on a silica gel 60 plate (Merck, Germany). The plate was developed twice using n-butyl alcohol/ethanol/water (5:3:2, *v*/*v*/*v*) and sprayed with a mixture of aniline diphenylamine and phosphoric acid, followed by heating at 90 °C for 10 min to detect sugar spots. The amounts of reducing sugars were quantified using the DNS method [42]. 

To gather more detailed information of XOSs, HPLC analysis was performed on an HPLC system (Thermo U-3000, Thermo Scientific, Waltham, MA, USA) equipped with a differential refractive index detector (RI-101, SHODEX) and a CarboPAC analytical column (250 mm × 2 mm, Dionex CarboPac PA10, Thermo Scientific, Waltham, MA, USA), which was maintained at 60 °C using deionized water as the mobile phase at a flow rate of 0.2 mL/min. A mixture of xylobiose and xylotriose was used as the standard. 

## 4. Conclusions

Xylanases are ubiquitous and diverse in the natural environment. Some of them have been used to decompose lignocellulosic materials in industry. Recently, cellulase-free xylanases have received great attention due to their environmental friendliness. Actinomycetes are particularly interesting producers of xylanases from an industrial point of view. In this paper, an endo-β-1,4-xylanase gene (*xyn*ST7) from *Streptomyces* sp. T7 was identified through genome sequence alignment and enzyme assay. XynST7 as a member of the glycoside hydrolase family 10 exhibited good pH adaptability and thermostability, while XynST7 had no cellulase activity, and it was able to efficiently hydrolyze xylans extracted from raw corncobs to produce xylo-oligosaccharides (XOS) without xylose, mainly being xylobiose and xylotriose. Therefore, XynST7 is of great value for further research and application developments.

## Figures and Tables

**Figure 1 molecules-27-02516-f001:**
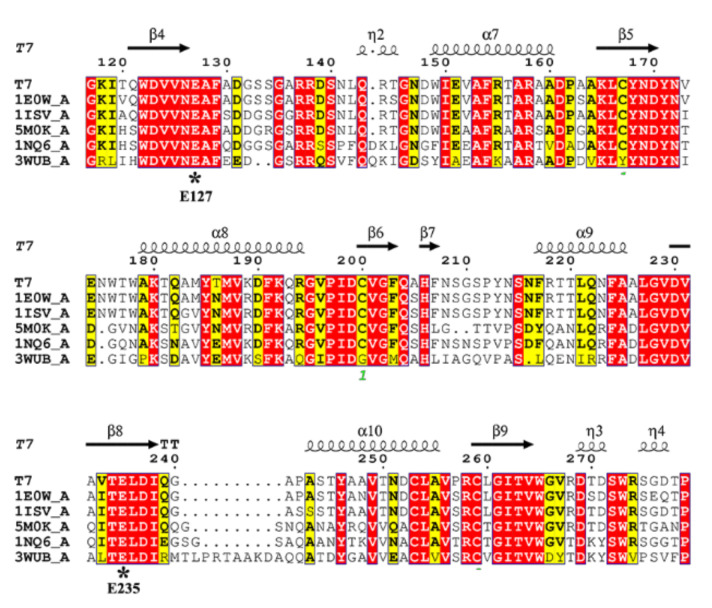
Amino acid sequence alignments of endo-β-1,4-xylanases using the online ClustalW tool. The species names are listed in the sequence alignment as follows: T7, XynST7 from *Streptomyces* sp. T7; IE0W_A, *S. lividans*; 1ISV_A, *S. olivaceoviridis* E-86; 5M0K_A, *Cellulomonas flavigena*; 1NQ6_A, *S. halstedii* JM8; 3WUB_A, *Streptomyces* sp. 9. The numbering of the amino acids starts at the N terminus of all sequences. Identical residues are shaded in red, while conserved residues are shaded in yellow. Arrows stand for β-strands, while helical curves denote α-helixes.

**Figure 2 molecules-27-02516-f002:**
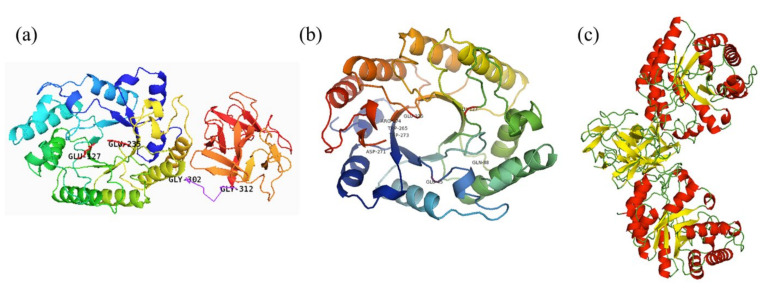
The overall (**a**), TIM-barrel (**b**), and dimeric (**c**) structures of endo-β-1,4-xylanase from *Streptomyces* sp. T7 (XynST7). The conserved catalytic sites are labeled.

**Figure 3 molecules-27-02516-f003:**
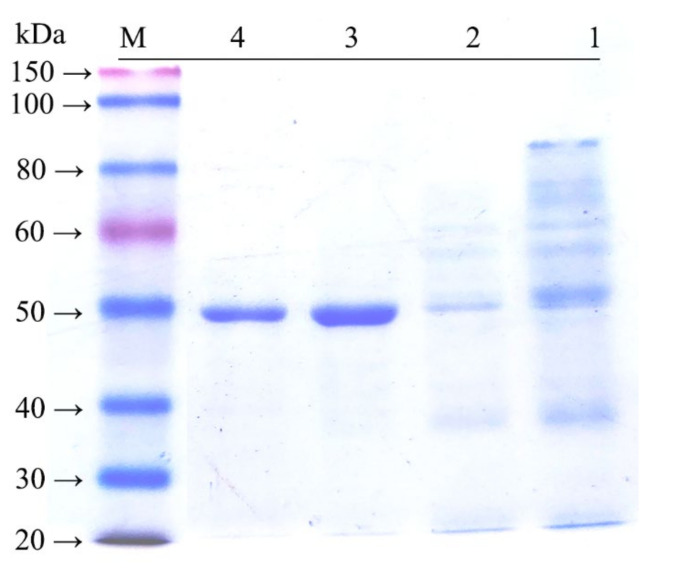
SDS-PAGE analysis of endo-β-1,4-xylanase expressed in *E. coli* BL21 (DE3). Lanes: M, protein molecular weight marker; 1, crude enzyme after ultrasonication; 2, Eluted fraction with 10 mM imidazole buffer; 3, Eluted fraction with 100 mM imidazole buffer; 4, purified fraction after ultrafiltration.

**Figure 4 molecules-27-02516-f004:**
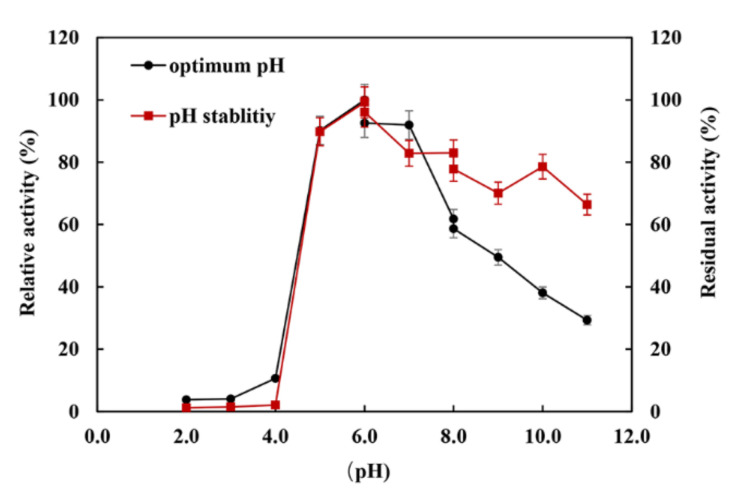
Effects of pH on the enzyme activity and stability of XynST7.

**Figure 5 molecules-27-02516-f005:**
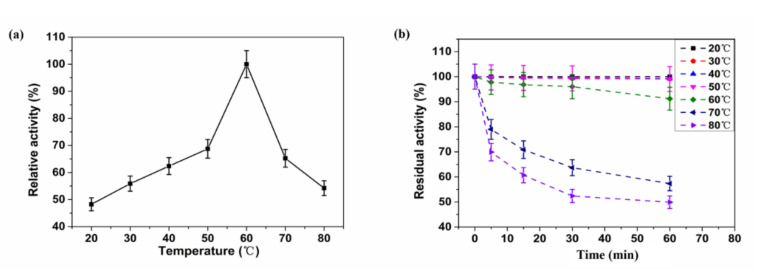
Effects of temperature on enzyme activity and stability of XynST7: (**a**) optimum temperature of XynST, (**b**) thermostability of XynST.

**Figure 6 molecules-27-02516-f006:**
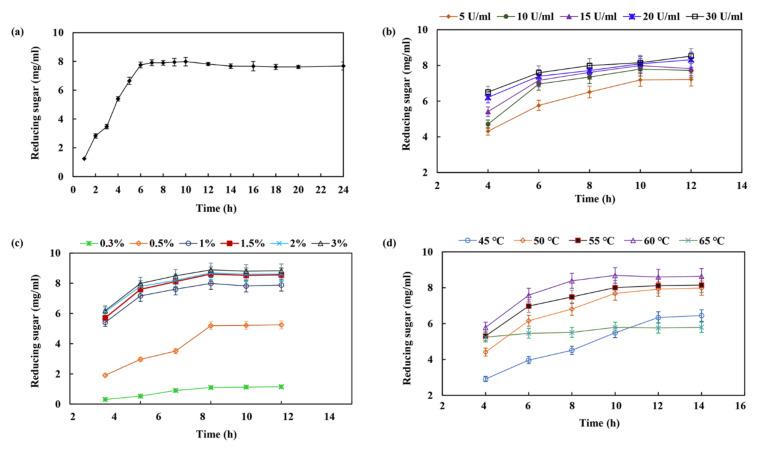
Effects of various factors on the production of XOS from corncob xylan via enzymatic hydrolysis. The factors are as follows: (**a**) time course; (**b**) enzyme dose (xylan, U/mL); (**c**) substrate concentration (corncob xylan, mg/mL); (**d**) temperature (°C).

**Figure 7 molecules-27-02516-f007:**
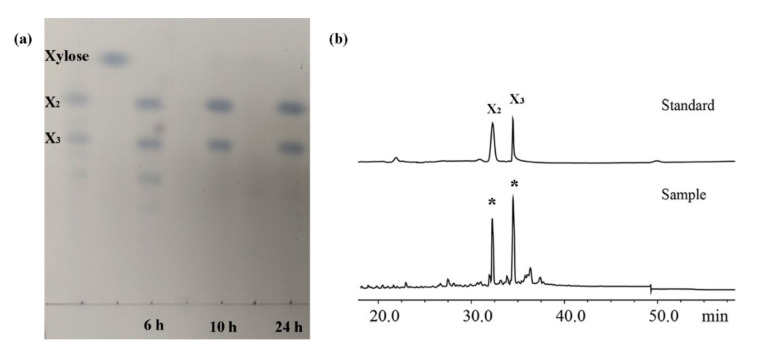
TLC (**a**) and HPLC (**b**) analyses of hydrolysate products. X2 and X3 represent xylobiose and xylotriose, respectively. Note: 6 h, 10 h, and 24 h represent hydrolysate products after 6, 10, and 24 h.

**Table 1 molecules-27-02516-t001:** Effects of metal ions and chemical reagents on the activity of XynST7.

Metal Ions/Chemical Reagents	Relative Activity (%)		
	5 mM	2.5 mM	1 mM
KCl	80.5 ± 0.59	87.2 ± 0.60	102.9 ± 0.62
NaCl	92.9 ± 0.73	98.9 ± 0.40	99.7 ± 0.40
Ni^2+^	95.6 ± 0.49	97.7 ± 0.38	98.0 ± 0.17
CaCl2	86.7 ± 0.60	92.0 ± 0.39	95.1 ± 0.20
MnSO4	110.4 ± 1.70	109.9 ± 0.31	107.4 ± 0.29
FeSO4	43.9 ± 0.6	49.9 ± 0.68	85.9 ± 0.38
CoCl2	96.5 ± 0.73	99.5 ± 0.42	104.1 ± 0.38
MgCl2	86.2 ± 2.27	88.0 ± 1.86	101.2 ± 0.24
FeCl3	23.7 ± 2.97	25.8 ± 1.51	27.2 ± 0.28
EDTA	24.8 ± 0.92	64.6 ± 1.04	82.5 ± 0.46
SDS	23.2 ± 2.64	35.3 ± 1.32	41.7 ± 0.25

**Table 2 molecules-27-02516-t002:** Substrate specificity and kinetic properties of XynST7.

Substrate	Enzyme Assay		Kinetic Parameter		
	Enzyme	Activity Units (U/mL)	*K*_m_ (mg/mL)	*V*_max_ (U/mg mL)	*k*_cat_/*K*_m_ (s^−1^·mg^−1^)
*p*NPA	α-arabinofuranosidase	<0.05	NA	NA	NA
*p*NPX	β-xylosidase	<0.05	NA	NA	NA
*p*NPGα	α-glucosidase	<0.05	NA	NA	NA
*p*NPGβ	β-glucosidase	<0.05	NA	NA	NA
CMC	Cellulase	<0.5	NA	NA	NA
Avicel	Cellulase	<0.5	NA	NA	NA
Corncob xylan	Xylanase	235.8 ± 3.1	2.33 ± 0.15	678.41 ± 1.35	58.23
Birchwood xylan	Xylanase	203.3 ± 2.8	2.78 ± 0.21	596.54 ± 2.12	42.91
Oat spelt xylan	Xylanase	193.3 ± 2.5	3.12 ± 0.11	545.33 ±1.18	34.96
Beechwood xylan	Xylanase	188.6 ± 3.6	3.54 ± 0.13	495.23 ± 1.87	27.98

NA represents no test; *K*_m_ is the substrate affinity constant; *k*_cat_ is the turnover of the enzymatic reaction; *k*_cat_/*K*_m_ represents the catalytic efficiency.

**Table 3 molecules-27-02516-t003:** Properties and XOS production of XynST7 and other GH10 xylanases.

Sources	pH	Thermostability	XOS Production		References
	Stability	(°C)	Main Type	Xylose	
*Penicillium funiculosum*	4.0–5.5	≤70	X2	–	[38]
*Jonesia denitrificans*	6.0–9.0	≤55	X4, X5, X6	*	[39]
*Thermobacillus composti*	5.0–9.0	≤65	X2, X3	*	[40]
*Paludibacter propionicigenes*	4.0–9.0	<50	X2, X3	*	[41]
*Streptomyces* sp. B6	5.0–9.0	≤70	X2, X3, X4	**	[42]
*Aspergillus fumigatu*s var. niveus	4.5–7.0	≤70	X2, X3	–	[43]
*Streptomyces* sp. T7	5.0–11.0	≤80	X2, X3	–	this paper

Note: – no detection; * a small amount; ** a large amount.

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
