# Peer review of "Identification and Characterization of a Novel Endo-β-1,4-Xylanase from Streptomyces sp. T7 and Its Application in Xylo-Oligosaccharide Production"

_molecules, 2022, doi:10.3390/molecules27082516_

Round 1
Reviewer 1 Report
The manuscript by Li et al. presents the identification and characterization of an endo-beta-1,4-xylanase from Streptomyces. Firstly, the authors isolated a Streptomyces strain that can degrade xylan, a major component of the plant cell wall. Then, they sequenced the strain and identified the gene encoding an endo-beta-1,4-xylanase. Expression and purification of the gene product afforded a 47 kDa protein with endo-beta-1,4-xylanase activity. And this protein has maximum hydrolysis for corncob xylan. At last, the authors analyzed the products, which mainly consisted of xylobiose and xylotriose, indicating this endo-beta-1,4-xylanase is a candidate for various industrial applications.
Endo-beta-1,4-xylanases have been well characterized, and several structures have been reported and deposited. The protein charactered in this work is among the normal endo-beta-1,4-xylanases, mostly isolated from Streptomyces, without exceptional properties. Hence, this work is not novel. The authors conducted relatively comprehensive experiments to characterize this protein. As the authors claim, this protein may be useful in industrial applications.
Major points:
- How did the authors identify the endo-beta-1,4-xylanase (xynST7) in the sequenced genome? If the identification is based on sequence alignment, are there other endo-beta-1,4-xylanase genes in the genome? If yes, why did the authors continue with this specific gene?
- The manuscript is roughly prepared: all the figure legends are missing; there are no tables in the paper, but the authors mentioned "Table 2"; the paper needs a conclusion section, etc.
- Analyzing the homology model (Line 101 to 120) is meaningless, the authors may want to reduce this content.
Minor Points:
- Italicize all gene names, including xynST7.
- Line 56, "A" should be "a"
- Line57 to 59, actinomycetes are bacteria. They are not coordinate.
- Line 59, Italicize the specie names.
- Add citations that characterize proteins in Line 87 to 89.
Author Response
Please see the attachment, thank you.

Reviewer 2 Report
Li and collaborators isolated and identified a xylanase producing bacterial strain (Streptomyces sp. T7) from soil samples.
The manuscript is interesting but its discussion is poor and must be improved. Aspects such as potential applications for the xylanase, its advantages in comparison with others, advantages of the bacterial strain, etc; must be addressed.
Author Response
Please see the attachment, thank you.

Reviewer 3 Report
Authors reprort the identification and recombinant expression of a novel xylanase.
The study is well planned and rpresented.
Few comments to address:
- it might be meaningful to present (as a table) the sequence ismilarity of theis new xylanases with well studied ones. Are its features really unexpected?
- figure captions are missing.
- please comment on the possible glycosylation of the enzyme from the original host.
Author Response
Please see the attachment, thank you.

Round 2
Reviewer 1 Report
The authors have addressed my concerns, paper can be published after text editing.
Author Response
Dear Reviewers,
We quite appreciate your favorable consideration and insightful comments again. We have revised manuscript “molecules-1618562” based on your comments. We hope this revision can make our paper more acceptable. The reviewers’ comments were quite helpful for us, and we carried out a detailed revision for our paper in a point-by-point way according to the reviewers’ comments.
Thank you very much. English language and style have been corrected (attached).
